# Rapid high-throughput isolation and purification of chicken myoblasts based on deterministic lateral displacement microfluidic chips

**Lihong Gu** [1]*, **Hongju Liu**[2], **Long Wang**[2], **Haokai Fan**[1], **Xinli Zheng**[1], **Tieshan Xu**[3], **Qicheng Jiang**[1], **Teng Zhou**[2], **Liuyong Shi** [2]*

**1** Institute of Animal Science and Veterinary Medicine, Hainan Academy of Agricultural Sciences, Haikou, China, **2** School of Mechanical and Electrical Engineering, Hainan University, Haikou, China, **3** Tropical Crops Genetic Resources Institute, Chinese Academy of Tropical Agricultural Sciences, Haikou, China

\* gulihong@hnaas.org.cn (LG); shiliuyong@hainanu.edu.cn (LS)

## Abstract

Myoblasts are defined as stem cells containing skeletal muscle cell precursors. However, there are some challenges associated with the purification of myoblast samples, including long culture times and ease of bacterial contamination. In this study, we propose a microfluidic myoblast cell enrichment and purification platform based on the principle of deterministic lateral displacement (DLD). To achieve this, we designed a DLD chip with three outlets and tested it on 11-day-old (E11) *Wenchang chicken* pectoral muscle tissue. A cell suspension was prepared using the collagenase method, pretreated, and then passed into the designed DLD chip for myoblast enrichment and purification. In this study, the number of myoblasts and the diameter of myoblasts increased slowly before E9, and the diameter of myofibers decreased and the number of myofibers increased rapidly after E9. The period when the muscle fibers are most numerous is on the E12, and the period when the diameter of the muscle fibers begins to increase again after reaching its lowest point is also on the E12. After E12, the diameter of the muscle fibers increased and the number of muscle fibers decreased. At E12, myoblasts clustered and fused, and the proliferation of myoblasts was greatly reduced. E12 is both intact myoblasts and the most vigorous proliferation period, so the best time to determine isolation is E12. We attained a myoblast cell recovery rate of 80%, a target outlet collection purity of 99%, and a chip throughput of 50 μ m/min. In this paper, we innovate chips design according to specific geometries and functions for Wenchang chicken pectoral muscle tissue, so as to optimize the isolation and purification process of myoblasts. This study provides a novel and effective method for the isolation and purification of skeletal muscle myoblasts.

**Data Availability Statement:** All relevant data are within the manuscript and its Supporting Information files.

**Funding:** This research is funded by the National Key R&D Program of China (Grant No.2021YFD1300100), the Key R&D projects in Hainan Province (ZDYF2023XDNY036, ZDYF2022SHFZ033 and ZDYF2022SHFZ301), the Project of Wenchang city Wenchang Chickens Research Institute (WWXM20230301), and the Animal Branch of the Germ plasm Bank of Wild Species, Chinese Academy of Sciences. The funders had no role in study design, data collection and analysis, decision to publish, or preparation of the manuscript.

**Competing interests:** The authors have declared that no competing interests exist.

## 1. Introduction

China is the second-largest producer and consumer of chicken meat worldwide and the broiler industry is an important component of agricultural productivity and the rural economy in the country [1]. The robust development of the broiler industry has led to improvements in the dietary structure of urban and rural residents, providing important nutrients, such as protein, and has also played a pivotal role in the international meat trade [2]. There is an increasing demand for quality chicken, which is accelerating the study of muscle development. Mature muscles are made up of muscle fibers, which are formed through the fusion of myoblasts. Therefore, a deeper understanding of skeletal muscle myoblasts can contribute to the improvement of muscle production and quality as well as to muscle repair and regeneration.

Mesoderm-derived muscle progenitor cells refer to populations of cells that differentiate from the mesoderm during embryonic development and have the potential to differentiate into muscle cells. The mesoderm is one of the three germ layers of the embryo and is responsible for the generation of a variety of tissues and organs, including bones, blood vessels, and muscles. Muscle progenitor cells are gradually differentiated into myoblasts under specific signals and gene regulation (e.g., Myf5 and MyoD), and these myoblasts eventually fuse to form myofibers. Mesoderm-derived muscle progenitor cells play a vital role in the formation and regeneration of muscle tissue.

Myoblasts are defined as stem cells containing skeletal muscle cell precursors [3]. They not only play a key role in the repair and regeneration of muscle fibers following trauma but also have multi-differentiation potential [4]. Given these characteristics, myoblasts have attracted increasing attention from researchers for their potential for application in tissue engineering [5]. Myoblast cell isolation and purification underlies myoblast research. Traditional methods for isolating and purifying myoblasts from skeletal muscle include differential adherence assays [6, 7], density gradient centrifugation [8], and flow cytometry [9]. Although these methods are still in use, they all have shortcomings, including the length of culture in differential adherence assays, complexity in density gradient centrifugation, and high equipment costs in flow cytometry.

Microfluidics is a novel cell separation technology that relies on the physical and chemical properties of fluids in micro- and nano-fields to complete the process of separation, purification, and detection. This technique not only compensates for the shortcomings of traditional cell isolation techniques, but also has the advantages of miniaturization, low sample consumption, fast analysis, and ease of integration [10]. Microfluidic technology has been widely used for the isolation and detection of a variety of cell types [11–13]. Microfluidic separation and purification technologies are generally divided into active and passive methods. Active separation involves achieving high-precision particle separation using light, sound, electricity, magnetism, or other means, while passive separation involves the use of the specific structure of the chip to sort the particles in a fluid. Among the microfluidic devices, the active chip design is more complex and cumbersome than the passive one; accordingly, passive separation is the preferred method. Deterministic lateral displacement (DLD) is a widely used technique in passive separation and has the advantages of simple structure, ease of processing, and high separation ability. It is extensively used for the isolation of a variety of cell types [14–17], especially human blood and tumor cells [18–20].

DLD is a passive separation technology that achieves the separation of particles through different outlets based on the different movement trajectories of particles of different sizes in a microcolumn array. The types of micropillars generally include those of varying regular and irregular shapes, the most common ones being circles [21–23] and triangles [19, 24].

In the DLD array, the streamlines divide the laminar flow into different regions. The model shown in **Fig 1** is divided into three laminar flows, each with the same flow rate. Laminar flow

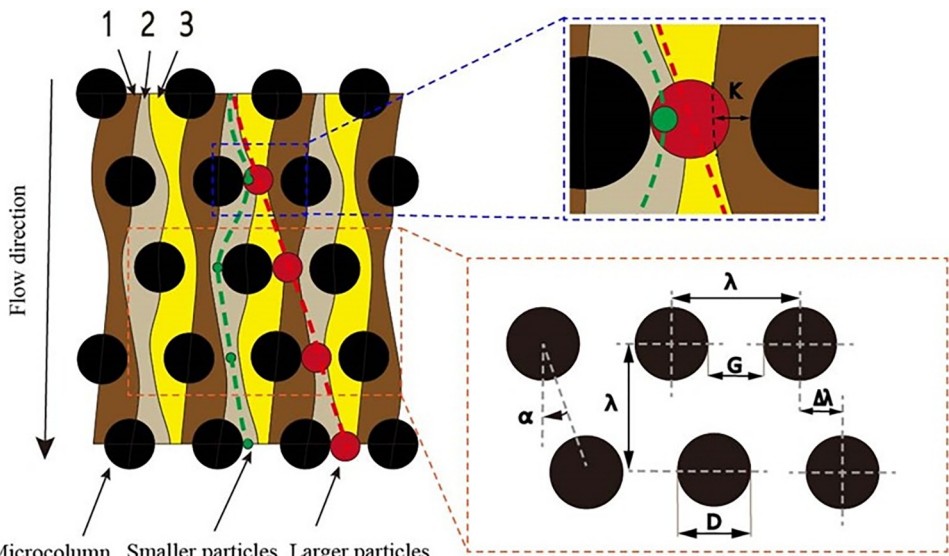

**Fig 1. Deterministic lateral displacement (DLD): Basic schematic.** Note: The black dots are the micropillars; the curves between the micropillars represent the streamlines; the red and green dots are the larger and smaller particles, respectively; 1, 2, 3 are laminar flows; the arrows represent the fluid flow direction; $k$ is the laminar flow width; $\lambda$ represents the transverse gap between the pillars as well as the longitudinal gap; $\Delta\lambda$ represents the offset of the right pillar relative to the adjacent left pillar in the vertical direction; $\alpha$ indicates the angle of the right pillar relative to the left pillar; G is the interval between the pillars, and D is the diameter of the pillars.

describes how fluids (liquids or gases) move smoothly and orderly in layers or sheets, with each layer sliding past the adjacent one without mixing much. Imagine water flowing gently in a straight pipe where the layers of water move parallel to each other, like sheets of paper stacked neatly. When the particle radius is less than k, a particle will move along the original laminar flow (laminar flow 1) when passing through the microcolumn gap, that is, it will move along the trajectory of the streamline (green dotted line), and the overall trajectory will be a zigzag trajectory. When the particle radius is greater than k, the particle will cross the original flow layer and move along the laminar flow in a 2-3-1 direction when traversing the laminar flow in the microcolumn gap, which not only produces vertical flow, but lateral movement, that is, the particle will move along the collision trajectory, change to a different flow layer, and the overall trajectory will be the collision trajectory. Particle trajectories refer to the paths that particles follow as they move through a fluid. Similarly, when myoblast cells are introduced into the chip, large cell clumps and fibroblasts that exceed the critical radius collide with the micropillar and are displaced sideways and deflected to one side; meanwhile, myoblasts smaller than the critical flow diameter through the array according to their original flow direction, and can be separated via different outlets, resulting in their isolation and purification.

The incubation time of *Wenchang chickens* is 21 days. During this time, mesoderm-derived muscle progenitor cells are selected to form myoblasts, which then differentiate and fuse to form myotubes, which themselves combine to form mature muscle fibers. This means that myoblasts must be isolated before they differentiate and form myotubes. Therefore, in this study, we conducted a preliminary study on pectoral muscle tissue of *Wenchang chickens* between embryonic day 7 (E7) and E14, and found that the myoblast cell proliferation rate was fastest on E12, at which stage differentiation had not yet begun. This was the best period for myoblast isolation. Compared with round micropillars, triangular micropillars are less sensitive to flow velocity, can carry a larger flux, and have higher sorting accuracy. Our results

represent a novel means for isolating animal cells, especially avian ones, in a rapid and high-throughput manner, with a high recovery rate, and without affecting cell viability.

## 2. Materials and methods

### 2.1 Animal study protocols

The animal study protocol was approved by the National Key R&D Program of China (Grant No.2021YFD1300100), the Key R&D projects in Hainan Province (ZDYF2023XDNY036, ZDYF2022SHFZ033 and ZDYF2022SHFZ301), the Project of Wenchang city Wenchang Chickens Research Institute (WWXM20230301), and the Animal Branch of the Germ plasm Bank of Wild Species, Chinese Academy of Sciences.

### 2.2 Tissue sectioning and hematoxylin and eosin staining

Pectoral muscle samples obtained from E7 to E15 chicks were rinsed with tap water and were dehydrated twice by passing through a graded alcohol series (75% alcohol for 4 h, 85% alcohol for 4 h, 95% alcohol overnight, and 100% alcohol for 2 h). The dehydrated tissue specimens were treated with xylene three times, paraffin-embedded, trimmed, and cut into thin, 4-μm-thick sections. The sections were then placed in a water bath at approximately 40°C for 5 min, transferred to glass slides, air-dried for 30 min, and baked at 45°C overnight. The sections were deparaffinized with two concentrations of xylene, 10 min each step, dehydrated twice with 100% ethanol (2 × 3 min) and once with 95% and 80% ethanol (1 min each step), and finally rinsed with distilled water for 5 min. The sections were then stained with hematoxylin and eosin (H&E).

The sections were observed under an inverted microscope (Nikon.DFC450C Germany) at different magnifications. A total of 125 images were captured for each sample (and each individual selected 5 better films, each film took 5 clear fields of view, take 5 photos per field of view). Image-Pro Plus 6.0 software (USA) was used to measure myofiber diameter and the number of myofibers per unit area in each image.

### 2.3 Preparation of cell suspensions and reagents and equipment

Collagenase digestion was performed as previously described [25].

The animal samples were isolated from *Wenchang chicken* embryos and eggs obtained from Hainan Chuan Wei Wenchang Industry Co., Ltd. Type I collagenase was purchased from Solarbio (China); trypsin and DMEM, Low Glucose, were purchased from Sigma; fetal bovine serum, FITC-labeled goat anti-mouse IgG, PBST buffer (500 mL), and 0.5% Triton X-100 (100 mL) were purchased from Biyuntian; anti-PAX7 antibody was obtained from Santa Cruz Biotechnology; PBS and penicillin/streptomycin were from Gibco; 4% paraformaldehyde (100 mL) and 1% BSA (100 mL) were from Biosharp, Inc. The test equipment included an inverted microscope (OLYMPUS IX73), a high-speed camera (CDD), a dual-channel high-precision syringe pump (Cchippump01-BD, SMIC Qiheng), and a desktop computer.

### 2.4 Chip design and processing and DLD separation platform

Critical dimension $D_C$ was defined by Eq 1.1:

$$D_C = 2k \tag{1.1}$$

where $k$ is the laminar flow width.

The parameter relationship of the micropillar array is shown in Eq 1.2:

$$\varepsilon = \Delta\lambda/\lambda = 1/\mathrm{N} = \tan\alpha \tag{1.2}$$

where $N$ is the period of the micropillar, which represents the center distance between longitudinal adjacent pillars, $\lambda$ is the declination angle of the right pillar relative to the left pillar, $\Delta\lambda$ is the offset of the right pillar relative to the adjacent left pillar in the vertical direction, and $\varepsilon$ is the row transformation fraction, that is, the longitudinal offset of the left and right adjacent pillars; $\alpha$ is the offset angle.

The relationship between the critical value $Dc$ of the microcolumn and the microcolumn interval $G$, and the row transformation fraction $\varepsilon$ is shown in Eq 1.3:

$$\left[\frac{D_C}{G}\right]^3 - 3\left[\frac{D_C}{G}\right]^2 + 4\varepsilon = 0 \tag{1.3}$$

Triangular micro cartridges are insensitive to flow and have a high sorting accuracy [26]. Therefore, in this study, we used a triangular micropillar chip with the following parameters: the side length *(L)* of the triangular micropillar was 55 μm, the offset angle *($\alpha$)* of the micropillar was 5.7˚, the transverse center distance *($\lambda$)* was 65 μm, and the $D_C$ was 10 μm. The chips were prepared using a standard soft lithographic process.

The physical diagram of the DLD separation platform and chip is depicted in **Fig 2** (see **S1 Fig** in the S1 Appendix for the structural design diagram). The platform mainly included four parts—a syringe pump injection device, a DLD separation chip, a microscope and PC observation device, and a centrifuge tube collection device. The syringe pump injection device provided a stable flow rate, and the fluid flow rate could be adjusted from 0.1 to 999 μL/min using the syringe pump. The DLD chip had two inlets and three outlets (**Fig 3**), with inlets 1 and 2 leading to the sheath fluid and cell mixture, respectively. From bottom to top, on the right side of the chip, are outlet 1, outlet 2, and outlet 3, which were set up to facilitate the collection of purer myoblast cells. Pure myoblast cells are collected from outlet 1, myoblast cells and some of the rest of the cells are collected at outlet 2, and the rest of the cells, including clumped cells and fibroblasts, are collected from outlet 3. The image processing device included an inverted fluorescence microscope and a computer; the particle trajectories were imaged *via* the microscope and the images were transmitted to a computer.

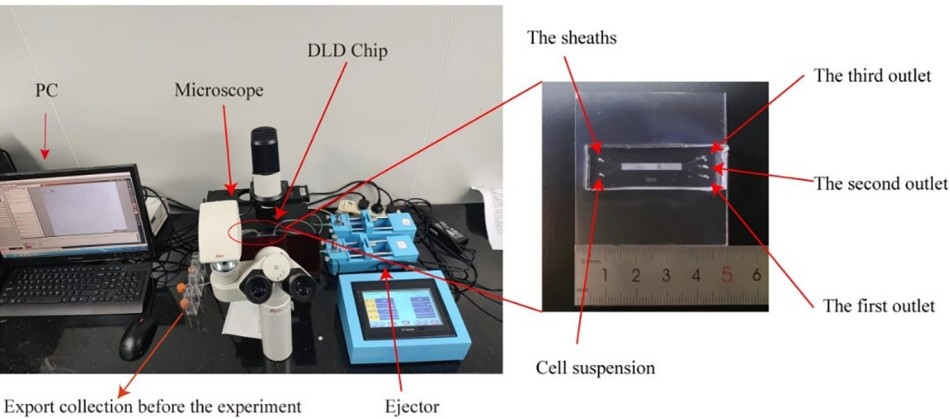

**Fig 2. Deterministic lateral displacement (DLD): Separation platform.**

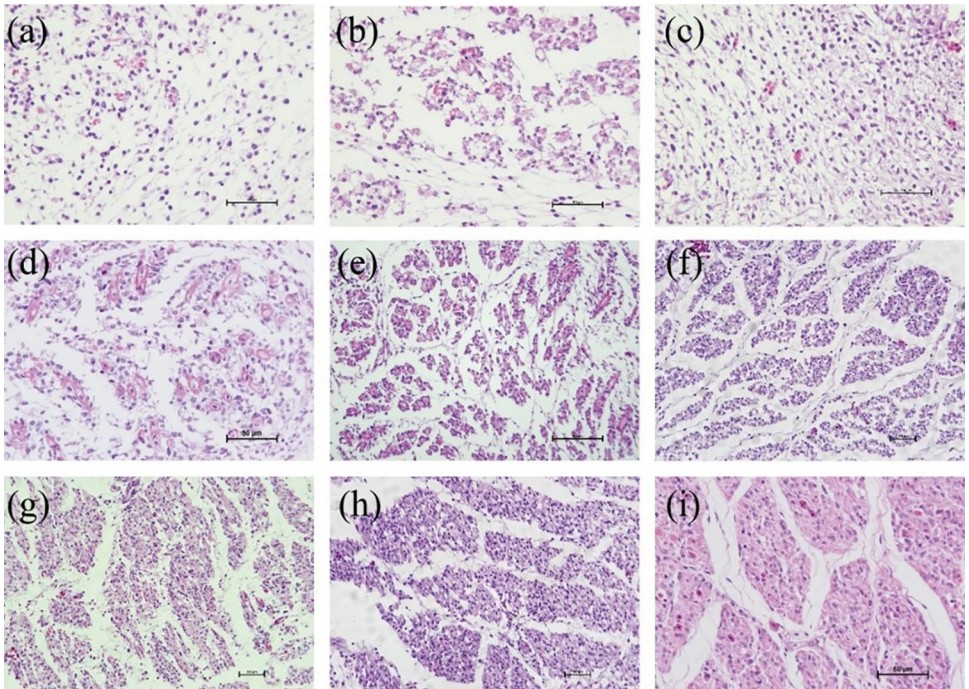

**Fig 3.** (a–i) The sections of *Wenchang chicken* pectoral muscle from embryonic day 7 to embryonic day 15.

## 2.5 Purification of myoblast cells by microfluidics

The cell suspension was washed first with deionized water before being passed through the DLD chip, and then with PBS; and was placed on the microscope stage for later use. Connect the syringe pump to the chip inlet with a tube, the inlet passes into the sheath fluid and the cell suspension, respectively, and the chip outlet is introduced into the collection cup with a tube. The flow rate of the syringe pumps was adjustable. The cells were collected and observed under a microscope. Outlet and inlet cells were counted using a cell counter and the recovery and purity of the outlet cells were calculated as follows:

Purity formula:

$$Purity = \frac{Number\ of\ single\ cells\ at\ an\ outlet}{Total\ number\ of\ cells\ at\ that\ outlet} \tag{1.4}$$

Recovery formula:

$$Recovery = \frac{The\ number\ of\ cells\ in\ one\ outlet}{The\ total\ number\ of\ cells\ at\ all\ three\ outlets} \tag{1.5}$$

## 2.6 Identification of myoblast cells

The reagents required for myoblast identification included Triton X-100 (Beyotime), murine anti-chicken PAX7 antibody (Santa Cruz Biotechnology), FITC-labeled goat anti-mouse IgG (H+L) antibody (Beyotime), and DAPI (Beyotime).

The procedures used for myoblast identification are described in [27].

## 3. Results

### 3.1 The development of *Wenchang chicken* pectoral muscle from embryonic day 7 to embryonic day 15

The development of *Wenchang chicken* pectoral muscle from E7 to E15 is shown in **Fig 3**. E7-E15 represents the period from the proliferative phase (producing more mononucleated muscle cells) to the fusion phase (formation of multinucleated myotubes). At E8 and E9, the cells showed significant proliferation and the number of nuclei increased. On E10, proliferation continues, cells begin to aggregate, and cell fusion is unclear. On E11, a blurred muscle fiber contour appears, and the cells begin to gather and gradually form the contour of the muscle bundle. From E12 to E15, myoblasts constantly accumulate and fuse to form intact fibrocytes.

E7–E15 represents the transition period from the proliferative stage (the generation of more mononuclear muscle fibers) to the fusion stage (the formation of multinucleated myotubes).

The changes in myofiber diameter and the number of myofibers per unit area in pectoral muscle tissue from E7 to E15 in *Wenchang chickens* are shown in **Figs 4 and 5**.

As seen in **Fig 4**, the diameter of the myofibers during the embryonic stage (E7–E15) in *Wenchang chickens* shows a trend of first slightly increasing, then decreasing, and subsequently significantly increasing; the inflection point of the small increase is on E9, and the inflection point of the large increase is on E13. **Fig 5** shows the overall change in the number of muscle fibers per square millimeter from E7–E15, which displays a trend of first increasing and then decreasing; the inflection point is on E12.

The production of pectoral myocytes in chickens during the embryonic stage involves proliferation followed by differentiation. The transition between the two is not obvious. After the proliferation of skeletal muscle myoblasts, some myofibroblasts enter the differentiation and

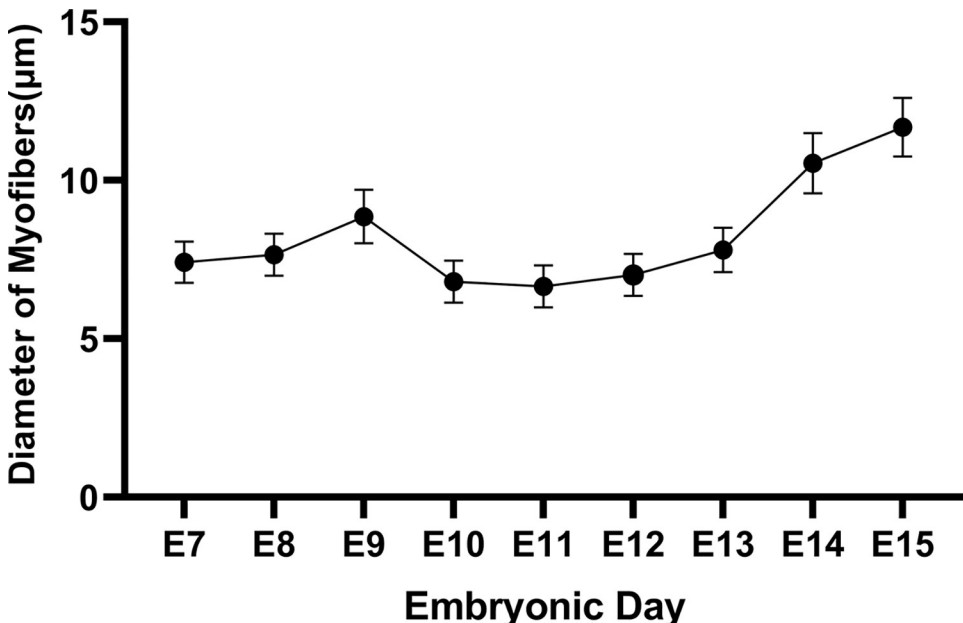

**Fig 4. Changes in the diameter of myofibers in pectoral muscle tissue of *Wenchang chicken* embryos from embryonic day 7 to embryonic day 15.** The inflection point of slight increase in myofiber diameter occurs at E9, while the inflection point of significant increase happens at E13.

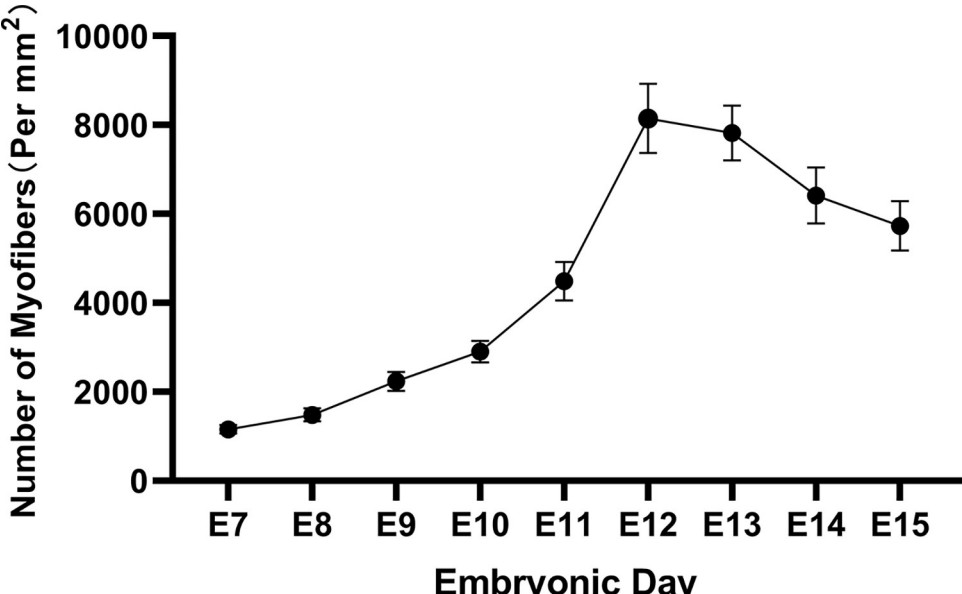

**Fig 5. Changes in the number of muscle fibers per unit area in pectoral muscle tissue of *Wenchang chicken* embryos from embryonic day 7 to embryonic day 15.** There is a trend of increasing fiber quantity followed by a decrease, with the inflection point occurring at E12.

fusion stage. Consequently, the stage at which skeletal muscle myoblast cells are isolated becomes particularly important. As demonstrated in **Figs 4 and 5**, the number of myofiber cells (mainly myoblast nuclei) and the diameter of myoblast cells increased slowly before E9; after E9, the diameter of myofibers (still mainly consisting of myoblast nuclei) decreased, while the number of myofibers increased rapidly. The number of myofibers (differentiated myoblast cells) peaked on E12, which is also when the diameter of myofibers increased again after the nadir. After E12, the diameter of the muscle fibers increased, whereas the number of muscle fibers decreased. On E12, myofibroblasts clustered and fused, and myoblast proliferation was greatly reduced. E12 represents the period when myoblast cells have completely formed and when proliferation is most vigorous; accordingly, E12 was determined to be the most appropriate time for myoblast isolation.

### 3.2 Cell characterization in cell suspensions

Cell size characterization in cell suspensions is a prerequisite for chip design and successful separation. Once pectoral muscle tissue isolated from chicken embryos of E12 had been passed through a 200-mesh and a 400-mesh cell sieve, the cell suspension mainly contained fibroblasts, clumped cells, and myoblasts (green, purple, and orange marks, respectively, in **Fig 6A**). The fibrocyte morphology was ellipsoidal, with symmetry along the axis, and could be construed as being oval; only the major and minor axes could be measured for size determination. The average length of the long axis was 45.78 μm and that of the short axis was 9.67 μm. The myoblast cells were spherical, almost round, and had a diameter of approximately 9.26 μm. According to these diameters, a suitable DLD chip was designed to isolate and purify the myoblast cells.

### 3.3 Trajectories of the different cells in the chip

The changes in the positions of the cells and the movement trajectories of cells of different sizes in the DLD chip were observed through a microscope, as shown in **Fig 7**. The

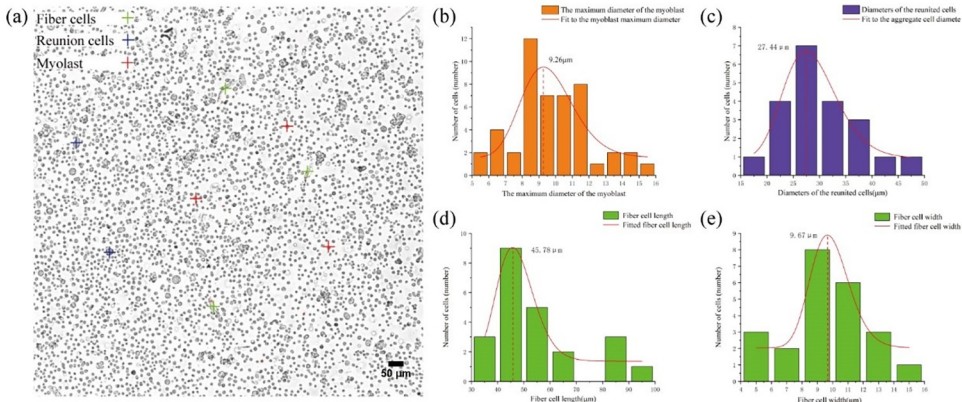

**Fig 6. Cell characterization following pectoral muscle dissociation.** (**a**) Microscopic images of three cell types. (**b**) Distribution of the maximum diameter of myoblast cells. (**c**) Distribution of the number of clumped cells with different diameters. (**d**) distribution of the number of fibroblasts of different lengths. (**e**) distribution of the number of fibroblasts of different widths.

agglomerated cells, with larger sizes (red circles), moved vertically from 35.2 mm (**Fig 7A**) to 36.4 mm (**Fig 7F**) from the bottom of the chip after passing through the eight columns of micropillars. The cell trajectory was a collision trajectory (white line). The larger cells moved along the micropillar tilt direction, and, after protracted horizontal movement, converged on the chip. The myoblast cells with a particle size smaller than the critical diameter (green circle) migrated through seven columns of micropillars and moved vertically from 13.8mm to 8.1mm from the bottom of the chip in a zigzag trajectory (white line); similarly, after moving some distance, the cells converged on the lower part of the chip. Consequently, the two cell types moved in opposite directions and eventually flowed out from different outlets, enabling the isolation and purification of cells of different sizes, i.e., smaller myoblast cells and larger cell clumps and fibroblasts. As can be deduced from **Fig 7**, the movement trajectories of cells of

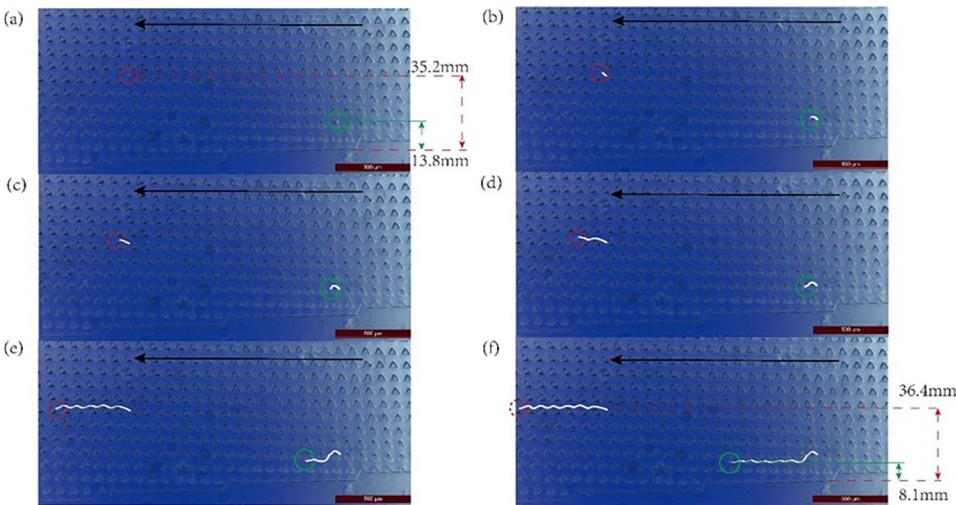

**Fig 7. Changes in the positions of cells and the trajectories of cells of different sizes at the microscopic level.** The black arrows indicate the direction of fluid flow; six frames (**a–f**) were selected to show the movement of cells of two different sizes in the fluid.

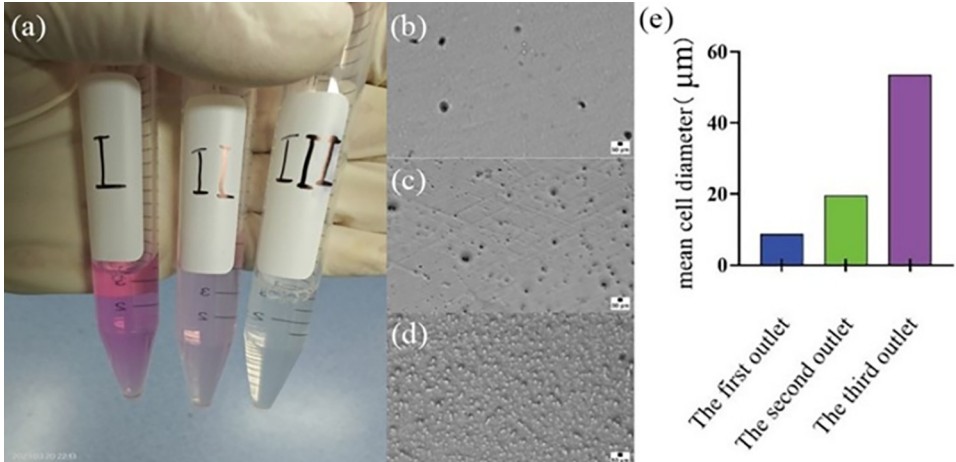

**Fig 8. Cell morphology and corresponding cell diameters at each outlet.** (**a**) Solutions collected from the three outlets. (**b**), (**c**), and (**d**) are cell morphologies at outlets 1, 2, and 3, respectively; (**e**) shows the mean cell diameter at each outlet.

different sizes at the microscopic level were consistent with the results predicted by DLD theory.

The cell suspensions collected from each outlet are shown in **Fig 8A**. The morphologies of cells at the different outlets are shown in **Fig 8B–8D**. The average cell diameter of each outlet is shown in **Fig 8E**. The solutions flowing out through the chip outlets were collected into different centrifuge tubes; the sizes and corresponding concentrations of the cells exiting the three outlets were characterized using a microscope and a cell counter, respectively. The average size of the separated cells differed markedly among the three outlets. The average size of the collected cells was 8.79 μm at the first outlet, 19.67 μm at the second outlet, and 53.51 μm at the third outlet. The purity of the target cells collected at each outlet was calculated using the purity formula (section 2.4) while the numbers of the different cell types at the three outlets were counted to calculate the corresponding recovery rate (**Fig 8**).

## 3.4 Cell fluorescence identification

The cells were incubated with FITC-conjugated goat anti-mouse IgG antibody (targeting PAX7), which labeled the cells green under excitation. Nuclei were counterstained with DAPI (blue). In fibroblasts and clumped cells, only DAPI staining was visible, whereas in myoblast cells, the nuclei were labeled both with DAPI (blue) and PAX7/FITC (green) (**Fig 9**).

Fluorescence identification of the sample cells from the three outlets showed that the round cells could be labeled with IgG antibodies at the same time as the cell membrane and DAPI nucleus, the color brightness was relatively large, and the color rendering was more vivid than that of other cells so that it could be identified by different fluorescence whether it was a myoblast cell. On the other hand, IgG antibodies are myoblast cell-specific markers, so round cells are regarded as myoblast cells; in contrast, long cells and larger cells can only be labeled with DAPI, and the color was lighter, so they cannot be labeled by IgG antibodies, so long cells and larger cells are regarded as fibroblasts and clump cells. The cells exiting each outlet were counted according to their morphology. The types and corresponding numbers of cells were quantified, and the purity and recovery rate of cells from each outlet were calculated (**Fig 8**). The cells collected from outlet 1 contained 99.9% myoblast cells, cells collected from outlet 2 contained 20% clumped cells and 80% myoblast cells, and those collected from outlet 3

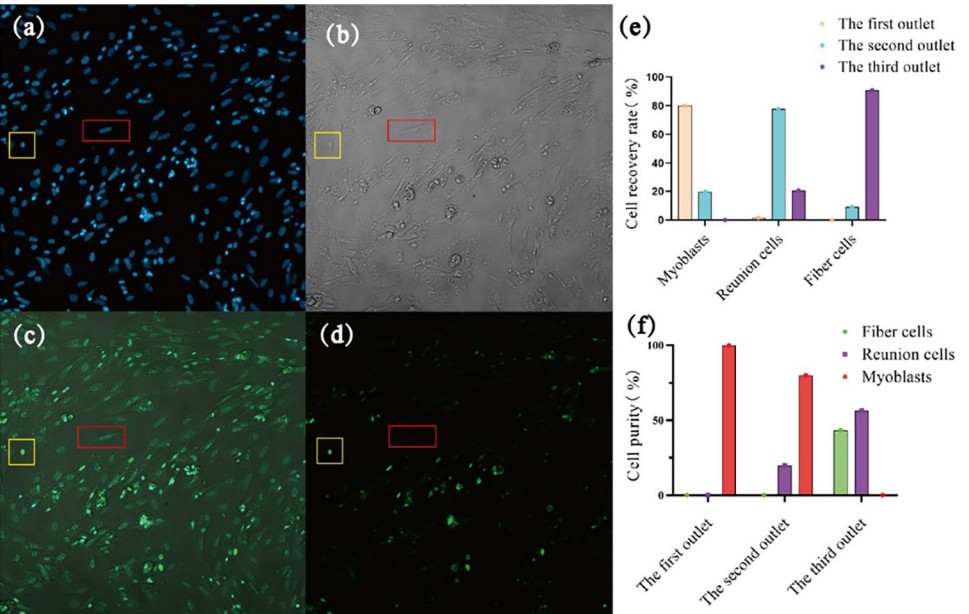

**Fig 9. Fluorescence-based identification of myoblast cells.** (**a**) Cell status under brightfield. (**b**) DAPI labeling of nuclei; the red box identifies a fibroblast and the yellow box identifies a myoblast. (**c**) Goat anti-mouse IgG antibody-labeled cells. (**d**) Merge of (**b–d**). (**e**) Histogram of the cell recovery rate. (**f**) Histogram of cell purity.

contained 56% fibroblasts and 44% clumped cells. Purer myoblasts were isolated from outlet 1, with a cell separation efficiency of up to 80%.

## 4. Discussion

Skeletal muscle myoblasts are important vectors for studying skeletal muscle myogenesis. In 2020 Subbiah V [28], constructed skeletal muscle organoids using myoblast cells, thereby providing a powerful *in vitro* 3D system for investigating muscle tissue morphogenesis and homeostasis. Traditional methods used for separating cells following enzymatic digestion of adherent cultures are complex and/or time-consuming. The microfluidic chip not only overcomes these shortcomings but also has two other unique advantages. First, microfluidic chips require fewer cells and thus can accommodate smaller sample sizes. Second, microfluidic chips can better mimic the complex cellular microenvironment when compared with traditional methods [29–31].in this study, we isolated and purified myoblast from the pectoral muscle of E12 embryonic stage using microfluidic chip method. The generation of pectoral myocytes during chicken embryogenesis involves two processes—proliferation and differentiation—and there is an organic transition between the two stages. Following the proliferation of skeletal muscle myoblast cells, some myofibroblasts enter the differentiation and fusion stage. Therefore, the stage at which skeletal muscle myoblast cells are isolated is particularly important. In this work, we found that the number of myoblasts (mainly myoblast nuclei), as well as the diameter of myoblast cells, increased slowly until E9. However, the diameter of myofibers (still comprising mainly myoblast nuclei) decreased, while the number of myofibers rapidly increased after E9. The number of myofibers (differentiated myoblast cells) peaked on E12, which was also when the diameter of the myofibers increased again after the nadir. After E12, the diameter of the muscle fibers increased, whereas the number of muscle fibers decreased. Additionally, myoblasts began to cluster and fuse, and myoblast proliferation was greatly

reduced. E12 presents both the intact myoblasts and the most vigorous proliferation period. Thus, we selected the pectoral muscle at E12 to isolate and purify myoblast.

Skeletal muscle myoblast cells are spherical, and the chip, based on the DLD principle, makes the myoblasts move in a deterministic lateral displacement direction, thus achieving separation. In this study, the isolation of skeletal muscle myoblast cells was performed entirely on a chip with a diameter of only 2.4 cm designed by the authors. The myoblasts with smaller sizes, agglomerated cells and fibroblasts with larger sizes can be separated completely, and the actual movement trajectories of cells with different particle sizes are consistent with the results predicted by DLD theory. Through the use of microfluidic technology, the purity of the collected myoblast cells reached 99.9% with the separation efficiency was as high as 80%, which indicates that microfluidic technology is effective in isolating and purifying the myoblast cells of *Wenchang chicken*. The deterministic lateral displacement method can be integrated with the traditional cell enrichment method, such as dielectrophoresis, removing some dead cells or cells with low viability, and trying other enrichment modes.

## 5. Conclusion

In this study, the principle of microfluidic DLD was applied to the isolation and purification of myoblast cells. E12 was selected as the stage to isolate the myoblast because of more myoblast cells existing at this stage. According to the sizes and shapes of cells, the microfluidic chip was designed. The purity of the collected myoblast cells reached 99.9%, while the separation efficiency was as high as 80%.

## Supporting information

**S1 Appendix. Some specific experimental procedures included tissue slice, cell suspension solution preparation, chip processing, manufacturing, deterministic lateral displacement (DLD) experiment and fluorescence identification are integrated in this file.**
(DOCX)

## Author Contributions

**Conceptualization:** Lihong Gu, Hongju Liu, Tieshan Xu.

**Data curation:** Lihong Gu, Hongju Liu, Tieshan Xu.

**Formal analysis:** Lihong Gu, Hongju Liu, Qicheng Jiang.

**Funding acquisition:** Lihong Gu.

**Investigation:** Lihong Gu, Long Wang, Qicheng Jiang.

**Methodology:** Long Wang, Qicheng Jiang, Teng Zhou.

**Project administration:** Long Wang, Haokai Fan, Teng Zhou.

**Resources:** Haokai Fan, Teng Zhou.

**Software:** Haokai Fan.

**Supervision:** Xinli Zheng, Liuyong Shi.

**Validation:** Xinli Zheng, Liuyong Shi.

**Visualization:** Xinli Zheng, Liuyong Shi.

**Writing – original draft:** Tieshan Xu.

**Writing – review & editing:** Tieshan Xu.

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
