## [Decision Letter · Decision Letter 0]

14 May 2024

PONE-D-24-09602Rapid High-Throughput Isolation and purification of Chicken Myoblasts Based on Deterministic Lateral Displacement Microfluidic ChipsPLOS ONE

Dear Dr. Gu,

Thank you for submitting your manuscript to PLOS ONE. After careful consideration, we feel that it has merit but does not fully meet PLOS ONE’s publication criteria as it currently stands. Therefore, we invite you to submit a revised version of the manuscript that addresses the points raised during the review process.

We look forward to receiving your revised manuscript.

Kind regards,

Zhiwen Luo

Academic Editor

PLOS ONE

https://journals.plos.org/plosone/s/file?id=ba62/PLOSOne_formatting_sample_title_authors_affiliations.pdf"

2.Thank you for stating the following financial disclosure: 

'This research is funded by the National Key R&D Program of China (Grant No.2021YFD1300100), the Key R&D projects in Hainan Province (ZDYF2023XDNY036, ZDYF2022SHFZ033 and ZDYF2022SHFZ301), the Project of Wenchang city Wenchang Chickens Research Institute (WWXM20230301), and the Animal Branch of the Germ plasm Bank of Wild Species, Chinese Academy of Sciences."

4.Your ethics statement should only appear in the Methods section of your manuscript. If your ethics statement is written in any section besides the Methods, please move it to the Methods section and delete it from any other section. Please ensure that your ethics statement is included in your manuscript, as the ethics statement entered into the online submission form will not be published alongside your manuscript. 

Reviewers' comments:

Reviewer's Responses to Questions

**Comments to the Author**

1. Is the manuscript technically sound, and do the data support the conclusions?

Reviewer #1: Yes

Reviewer #2: Yes

2. Has the statistical analysis been performed appropriately and rigorously? 

Reviewer #1: Yes

Reviewer #2: Yes

3. Have the authors made all data underlying the findings in their manuscript fully available?

Reviewer #1: Yes

Reviewer #2: Yes

4. Is the manuscript presented in an intelligible fashion and written in standard English?

Reviewer #1: No

Reviewer #2: Yes

5. Review Comments to the Author

Reviewer #1: 1. The introduction includes technical terms and concepts related to microfluidic technology and the isolation of myoblast cells. While it's important to include this detailed information, make sure to explain it clearly for readers who may not be familiar with the terms. Consider providing brief explanations or definitions for terms like "Deterministic Lateral Displacement" (DLD) and "Mesoderm-derived muscle progenitor cells" to enhance understanding.

2. Check for grammar errors, awkward phrasing, and language clarity in the text. Ensure that each sentence is clear and concise, and that the overall language is suitable for a scientific audience.

3. Detailed experimental protocols: Provide detailed protocols for each experimental procedure, including tissue sectioning, staining, cell suspension preparation, chip design, and DLD separation. This includes precise information on incubation times, temperatures, reagent concentrations, and specific equipment settings.

Reviewer #2: 1. The abstract should clearly state the significance and novelty of the proposed microfluidic method for myoblast cell isolation and purification.

2. Provide more detailed information on the design and fabrication process of the DLD chip, including specific dimensions and techniques used.

3. Include statistical analysis and data validation in the results section to ensure the robustness of the findings.

4. Add a brief section addressing the limitations of the method and suggesting avenues for further optimization and improvement.

6. PLOS authors have the option to publish the peer review history of their article (what does this mean?). If published, this will include your full peer review and any attached files.

Reviewer #1: **Yes: **Lei Sun

Reviewer #2: No

---

## [Author Response · Author response to Decision Letter 0]

26 Jun 2024

Please find our Response in the uploaded file.

---

## [Decision Letter · Decision Letter 1]

5 Jul 2024

PONE-D-24-09602R1Rapid High-Throughput Isolation and purification of Chicken Myoblasts Based on Deterministic Lateral Displacement Microfluidic ChipsPLOS ONE

Dear Dr. Gu,

Thank you for submitting your manuscript to PLOS ONE. After careful consideration, we feel that it has merit but does not fully meet PLOS ONE’s publication criteria as it currently stands. Therefore, we invite you to submit a revised version of the manuscript that addresses the points raised during the review process.

We look forward to receiving your revised manuscript.

Kind regards,

Zhiwen Luo

Academic Editor

PLOS ONE

Journal Requirements:

Additional Editor Comments:

Thank you for submitting your manuscript. We have now received the reviewers' feedback, and while they recognize the significance and potential impact of your work, they have also provided several suggestions for improvement.

To enhance the quality and clarity of your manuscript, we kindly request that you address the following points raised by the reviewers

Reviewers' comments:

Reviewer's Responses to Questions

**Comments to the Author**

1. If the authors have adequately addressed your comments raised in a previous round of review and you feel that this manuscript is now acceptable for publication, you may indicate that here to bypass the “Comments to the Author” section, enter your conflict of interest statement in the “Confidential to Editor” section, and submit your "Accept" recommendation.

Reviewer #1: (No Response)

2. Is the manuscript technically sound, and do the data support the conclusions?

Reviewer #1: Yes

3. Has the statistical analysis been performed appropriately and rigorously? 

Reviewer #1: Yes

4. Have the authors made all data underlying the findings in their manuscript fully available?

Reviewer #1: Yes

5. Is the manuscript presented in an intelligible fashion and written in standard English?

Reviewer #1: Yes

6. Review Comments to the Author

Reviewer #1: Reviewer's Comments

Thank you for your trust. Let me review this draft. This article still has some problems. The following are my review comments.

1. The explanation of the Deterministic Lateral Displacement (DLD) technique is highly technical and might be difficult for some readers to follow. Consider simplifying the language or providing additional explanatory text. Additionally, the reference to Fig. 1 could be enhanced by ensuring that the figure itself is clear and well-labeled, making it easier for readers to understand the concept without needing extensive background knowledge. Including a brief description of the key components (e.g., laminar flows, particle trajectories) in simpler terms would also be beneficial.

To improve readability and clarity, consider organizing the content into clearly labeled subsections with descriptive headings. For example, you can create distinct subsections for "Animal Study Protocols," "Tissue Sectioning and Staining," "Preparation of Cell Suspensions," "Chip Design and Processing," and "Purification of Myoblast Cells by Microfluidics." This will help readers navigate the methodology more easily.

3. Ensure that all critical parameters for reproducibility are included, such as the specific temperatures for paraffin embedding and the precise composition of buffers and solutions used.

4. Ensure that the descriptions of developmental stages and processes are clear and consistent throughout the text. For example, in the discussion of the transition from the proliferative stage to the fusion stage (E7–E15), clarify any ambiguous terms and consistently use scientific terminology to describe the stages of muscle development. This will help readers follow the progression of events without confusion.

5. Provide more detailed explanations for the figures referenced in the text. For example, when discussing Figures 4 and 5, include a brief description of the key observations from these figures and their significance to the study. This will help readers understand the context and relevance of the data presented in the figures without having to refer back and forth between the text and the figures.

6. Ensure that the statistical analysis and data presentation are robust and transparent. For example, describe the statistical methods used to analyze the changes in myofiber diameter and the number of myofibers per unit area (Figures 4 and 5). Include any relevant statistical metrics, such as p-values or confidence intervals, to support the significance of the observed trends. Additionally, consider presenting the data in a more visually accessible format, such as using bar graphs or line charts with error bars, to clearly convey the trends and variability in the data.

7. PLOS authors have the option to publish the peer review history of their article (what does this mean?). If published, this will include your full peer review and any attached files.

Reviewer #1: No

---

## [Author Response · Author response to Decision Letter 1]

2 Aug 2024

Please find our response in the attachment files.

---

## [Editor Report · Decision Letter 2]

8 Aug 2024

Rapid High-Throughput Isolation and purification of Chicken Myoblasts Based on Deterministic Lateral Displacement Microfluidic Chips

PONE-D-24-09602R2

Dear Dr. Gu,

We’re pleased to inform you that your manuscript has been judged scientifically suitable for publication and will be formally accepted for publication once it meets all outstanding technical requirements.

Kind regards,

Zhiwen Luo

Academic Editor

PLOS ONE
---

## [Editor Report · Acceptance letter]

15 Aug 2024

PONE-D-24-09602R2 

PLOS ONE

Dear Dr. Gu, 

I'm pleased to inform you that your manuscript has been deemed suitable for publication in PLOS ONE. Congratulations! Your manuscript is now being handed over to our production team.

Kind regards, 

on behalf of

Dr. Zhiwen Luo 

Academic Editor

PLOS ONE